# Long-Term Follow-Up of Mesothelioma Patients Treated with Dendritic Cell Therapy in Three Phase I/II Trials

**DOI:** 10.3390/vaccines9050525

**Published:** 2021-05-19

**Authors:** Daphne W. Dumoulin, Robin Cornelissen, Koen Bezemer, Sara J. Baart, Joachim G. J. V. Aerts

**Affiliations:** 1Department of Pulmonary Medicine, Erasmus MC Cancer Institute, 3015 GD Rotterdam, The Netherlands; r.cornelissen@erasmusmc.nl (R.C.); k.bezemer@erasmusmc.nl (K.B.); j.aerts@erasmusmc.nl (J.G.J.V.A.); 2Department of Biostatistics, Erasmus MC, 3015 GD Rotterdam, The Netherlands; s.baart@erasmusmc.nl

**Keywords:** malignant pleural mesothelioma, dendritic cells, vaccination, tumor lysate, immunotherapy

## Abstract

Background: Malignant pleural mesothelioma (MPM) is a fatal neoplasm with, if untreated, poor survival of approximately nine months from diagnosis. Until recently, phase II–III immunotherapy trials did not show any significant benefit. The lack of immunotherapy efficacy can be explained by the fact that mesothelioma is a tumor with an “immune desert” phenotype, meaning a non-inflamed tumor characterized by low T-cell infiltration. By administration of DCs, which were ex-vivo cultured, exposed to (tumor-associated) antigens, and subsequently activated, this “immune desert” phenotype might be turned into an “inflamed” phenotype. Three phase I/II studies have been performed and published using activated DCs, which support this concept. We here report on the long-term survival of patients treated with DCs in three phase I/II studies. Methods: Survival data of the phase I/II trials using DC therapy in MPM patients were obtained and subsequently analyzed. In the first two trials, DCs were loaded with autologous tumor lysate. In the third trial, DCs were loaded with allogeneic mesothelioma tumor cell line lysate. Results: In the three studies combined, 29 patients with MPM were treated with DC vaccination between 2006 and 2015. At data cut-off, the median OS was 27 months (95% CI: 21–47 months). OS at 2 years was 55.2% (95% CI: 39.7–76.6%), and OS at 5 years was 20.7% (95% CI: 10.1–42.2%). Conclusions: The long-term survival of DC therapy in MPM in these three trials is promising, which is the basis for the randomized phase II/III DENIM study. This DENIM study is currently enrolling, and the results of which have to be awaited for definite conclusions.

## 1. Introduction

Malignant pleural mesothelioma (MPM) is a fatal neoplasm of the pleural lining with poor survival of approximately nine months from diagnosis without treatment. Currently, treatment options are limited. Treatment with the combination of cisplatin with an anti-folate (pemetrexed or raltitrexed) resulted in a survival benefit of nearly three months [1,2]. Checkpoint inhibition therapy using pembrolizumab in the second line (PROMISE-MESO) improved response rate (RR) but did not improve progression-free survival (PFS) or overall survival (OS) compared to chemotherapy [3]. Recently, the phase 3 randomized CheckMate-743 trial showed an improvement in OS of four months in previously untreated malignant pleural mesothelioma with nivolumab (anti-programmed death-1 (anti-PD-1)) and ipilimumab (cytotoxic T-lymphocyte associated protein 4 (anti-CTLA-4)) compared to chemotherapy, leading to FDA approval [4]. The efficacy in the non-epithelioid subgroup was impressive, with an improvement in median OS of nearly ten months. For the epithelial subgroup, this improvement was, although significant, more modest—over 2 months, and long-term survival is unknown [5]. It should be noted, however, that both of these are prespecified subgroup analyses. This first positive trial regarding immunotherapy in MPM is in contrast with non-small cell lung cancer (NSCLC), where mono- immunotherapy targeting PD-(L)1 has become the standard of care based on many positive trials from 2015, showing long-term overall survival [6,7,8]. However, in mesothelioma, flattening of the curve is, to date, so far absent [5].

The lack of immunotherapy efficacy can be explained by the fact that mesothelioma is a tumor with an “immune desert” phenotype [9], meaning a non-inflamed tumor characterized by low T-cell infiltration. T-cells need to be activated by antigen-presenting cells; dendritic cells (DCs) are among the most potent antigen-presenting immune cells to activate these T-cells [10]. In mesothelioma, a low tumor mutational burden results in low numbers of tumor-associated antigens (TAA), leading to a challenging tumor recognition by DCs. Furthermore, the immunosuppressive tumor microenvironment, characterized by high numbers of immunosuppressive cells, such as M2 macrophages and regulatory T-cells, and high levels of immune-suppressive cytokines, such as vascular endothelial growth factor (VEGF), hinders the maturation of DCs and causes the absence of activated dendritic cells. [10]. Given that mature DCs are mandatory for an effective immune response, focusing on this step in the immune cycle could lead to a probably more viable treatment option for patients with mesothelioma [11]. DC’s can be maturated in several ways, either in vivo or ex vivo. A disadvantage of in vivo generation of DCs is that they may become inactivated by the immunosuppressive environment of the tumor. By generating DCs ex vivo, this can be prevented. DCs can be derived from monocytes, or they can be isolated in low levels from peripheral blood. For cancer immunotherapy, the ideal target for activating DCs would be a TAA that is exclusively expressed on all tumor cells without being present in normal tissues to prevent autoimmunity. Targeting multiple TAAs, as in tumor cell lysates, may overcome several disadvantages that can arise when using a single TAA [12]. For example, a single TAA may not be expressed on all tumor cells. Furthermore, when a single TAA is downregulated by the tumor, this will result in avoidance of immune detection. Polyvalent tumor cell lysates may be obtained either from autologous or allogeneic tumor cells. 

In mesothelioma, three phase I/II studies have been performed using activated DCs, which were cultured, activated, and exposed to antigens ex-vivo in order to overcome the problem of absent tumor recognition and absent maturation of DCs [13,14,15]. We hypothesize that treatment with activated dendritic cells could induce T-cell activation, preclude T-cell exhaustion, and increase long-term survival. Therefore, we collected the long-term survival of the patients treated with DCs in these three phase I/II studies.

## 2. Material and Methods

For this study, the survival data of three phase I/II trials were combined. In short, the studies were performed as follows:

In the first clinical trial (MM01), safety and immunological response by administering tumor lysate-pulsed dendritic cells were analyzed in patients with MPM [15]. Ten patients were treated with four cycles of standard chemotherapy followed by three vaccinations of mature DCs loaded with autologous tumor lysate and keyhole limpet hemocyanin (KLH) as a surrogate marker in 2-week intervals. Each vaccination consisted of 50 million DCs. The vaccinations were given 1/3 intradermally and 2/3 intravenously. In addition, peripheral blood mononuclear cells were drawn during the treatment in order to analyze immunological responses.

In a follow-up trial, the decrease in the number of regulatory T-cells and immunological responses in peripheral blood during treatment with autologous dendritic cell vaccination combined with low-dose cyclophosphamide were analyzed (MM02) [14]. Cyclophosphamide was added to reduce the number of Tregs. Ten patients were treated with four to six cycles of platinum and pemetrexed followed by DC vaccination combined with low-dose cyclophosphamide intermittently. In five of these patients, an additional pleurectomy/decortication was performed before DC vaccination. As in the first trial, the vaccinations of mature DCs were loaded with autologous tumor lysate and keyhole limpet hemocyanin (KLH) as a surrogate marker and were given 1/3 intradermally and 2/3 intravenously. The vaccinations were given with a 2-week interval 3 times, followed by revaccination after 6 and 12 months. Each vaccination consisted of 50 million DCs. Cyclophosphamide was administered daily starting 1 week prior to vaccination until the day of vaccination in a dose of 100 mg a day.

Using autologous tumor cell lysate, the number of patients eligible for vaccinations was limited due to insufficient amount and unsuitable tumor material, which hampers larger use of the DC vaccination. To overcome this challenge, a follow-up trial using an allogeneic mesothelioma tumor cell line lysate was performed (MM03). The aim of this trial was to investigate the safety and efficacy of allogeneic lysate-pulsed DC vaccination in mice and safety in humans [16]. Nine patients were treated with DC vaccinations consisting of autologous monocyte-derived DCs pulsed with tumor lysate originating from five different mesothelioma cell lines. DC vaccinations were given with a 2-week interval 3 times, followed by revaccination after 3 and 6 months. The vaccinations were administered 1/3 intradermally and 2/3 intravenously. The setup of this trial was a ±3 dose escalation safety analysis. Therefore, 3 patients received 10, 3 patients 25, and 3 patients 50 million DCs per vaccination.

### Statistical Analysis

In the current analysis, we collected the survival data of these three phase I/II trials. The median OS, 2-year OS, and 5-year OS, including 95% confidence intervals, were calculated based on the Kaplan–Meier curve. The analyses were repeated and stratified per study. R version 3.6.2 (R Core Team, 2019) was used for the statistical analysis.

## 3. Results

In the three studies combined, 29 patients with MPM were treated with DC vaccination between 2006 and 2015. Patient characteristics are summarized in Table 1. At data cut-off, the median OS was 27 months (95% confidence interval (CI): 21–47 months). OS at 2 years was 55.2% (95% CI: 39.7–76.6%), and OS at 5 years was 20.7% (95% CI: 10.1–42.2%).

Four patients are still alive, at respectively 71, 77, 114, and 128 months. The first two patients were treated with DC vaccinations containing allogeneic tumor lysate; the latter two patients were treated with autologous dendritic cell vaccination combined with low-dose cyclophosphamide; one patient has had a pleurectomy/decortication before the DC vaccination.

The survival analysis for the 3 separate trails is shown in Table 2. The first trial (MM01) showed a median OS of 15 months, a 2-year OS of 20% (95% CI 5.8–69.1%) and a 5-year OS of 10.0% (95% CI 1.6–64.2%). MM02 showed a median OS of 26 months, a 2-year OS of 60% (95% CI 36.2–99.5%), and a 5-year OS of 30.0% (95% CI 11.6–77.3%). MM03 showed a median OS of 31 months, a 2-year OS of 88.9% (95% CI 70.6–100%), and a 5-year OS of 22.0% (95% CI 6.6–75.4%), Table 2.

The all-grade toxicity of the 3 studies combined is presented in Table 3. DC vaccination therapy was well tolerated. Although adverse events were present in all patients, most were mild. The most common adverse events were a local skin reaction due to the intradermal injection and fever, which occurred 4–8 h from vaccination and resolved spontaneously within 24 h. One patient developed a cardiomyopathy 18 months after the DC vaccination therapy, which was defined by a left ventricular ejection fraction (LVEF) of 20%. This was probably due to the previous treatment with chemotherapy and could not directly be linked to the DC vaccination. In the following months, the LVEF improved to 50% and remained stable since then. 

## 4. Discussion

The long-term follow-up of MPM patients treated with DC vaccination in the three separate phase I/II trials shows a promising signal, with a 2-year OS of over 50% and a 5-year OS of over 20%. In addition, two patients are alive to date 10 years after treatment. In our opinion, these findings show the potency of DC vaccination therapy in the long-term activation of the immune system. Translational research performed in these studies did reveal that DC vaccination was able to induce a tumor-directed anti-T-cell response, the essential step for effective immunotherapy [13,17,18]. This opens the potential for combination immunotherapy with DC therapy as a backbone.

The three separate phase I/II trials had more or less similar study designs; in MM02, five patients underwent additional debulking surgery. With regard to immunotherapy, checkpoint inhibition therapy has shown to be more useful in patients with a low to modest tumor burden in melanoma and NSCLC patients [19,20,21]. A less prominent immunosuppressive tumor microenvironment can explain the improved effectiveness of immunotherapy in these studies in patients with a smaller tumor volume. Whether debulking surgery can cause an effect similar to an “earlier stage” cancer has yet to be determined. Also, whether a reduced tumor load is beneficial for DC vaccination therapy is a field of further research. As already mentioned, cyclophosphamide was added to this treatment regimen in order to reduce the number of Tregs, which was confirmed in peripheral blood analysis, and this strategy could be used in further research [22,23]. In addition, there might be a role of tumor mutational burden (TMB) as a biomarker for response to checkpoint inhibition [24]. Although TMB is generally low in MPM, it would be interesting to analyze the correlation between the height of TMB of the autologous tumor lysates and response to T-cell tumor infiltration after DC vaccination. Due to the time when this study was active, TMB was not analyzed and is therefore of high interest for future research. 

The use of autologous tumor material to load the DCs was labor intensive and cumbersome as in the majority of screened patients, not enough viable cells could be obtained to generate a lysate for DC loading. This resulted in only a minority of patients being eligible for participation in the first two studies. Therefore, an allogeneic tumor lysate was produced, which means that an “off-the-shelf” product is now readily available, which is also used in the current phase II/III trial. The long-term survival of the patients in the third study (MM03) is, in fact, not inferior to the patients treated in the previous trials. 

There are several limitations to this combined survival analysis. First, the primary outcomes of these phase I/II trials were safety and feasibility. OS was an exploratory outcome. Second, the three trials had a different setup; therefore, combining the outcomes should be approached with some caution. Third, there was no control group. The survival in mesothelioma patients is known to be variable, and long-term survivors do exist. However, when comparing survival data to historical data, the outcomes of patients treated with DC therapy are promising. Given that the patients treated in our three trials had to be non-progressive on the platinum-pemetrexed treatment, a comparison with the recently published NVALT-19 (Nederlandse Vereniging van Artsen voor Longziekten en Tuberculose) study is the most logical one. In the NVALT-19 study, patients were treated with switch-maintenance gemcitabine or best supportive care (BSC) if there was no progression on platinum-pemetrexed treatment [25]. In this study, OS at 2 years is approximately 25% in the gemcitabine group and 20% in the BSC group comparing to 55.2% OS for DC vaccination. Fourth, the treatment which was given after patients progressed after DC therapy varied over the years. Most patients were treated with second-line therapy in trials of named-patient programs, with checkpoint inhibition therapy being increasingly given over the years. In fact, eight of nine patients in the last trial were treated with immune checkpoint inhibition (ICI) therapy at some point in time. In theory, adding ICI therapy after DC therapy seems very logical. The T-cells are activated by the administration of the DC vaccines. In turn, these activated T-cells can be blocked by a PD-L1 expression of the tumor. ICI therapy would, therefore, be an ideal partner compound. If the seemingly improved survival is a result of this remains to be elucidated. 

DC vaccination therapy in MPM patients is currently being investigated in a large randomized phase II-III trial (NCT03610360) [26]. In the DENdritic cell Immunotherapy for Mesothelioma (DENIM) study, patients are randomized to allogenic tumor lysate-loaded DC vaccination therapy and BSC versus BSC alone after completion of 4–6 cycles of platinum-pemetrexed chemotherapy. The primary endpoint is OS. 230 patients will be enrolled at 6 sites in 5 countries. Enrolment is currently ongoing, and the trial is expected to complete enrolment in 2021. 

## 5. Conclusions

The long-term survival of mesothelioma patients after DC therapy in three phase I/II trials is promising. Results of the randomized phase II/III DENIM study, which is currently enrolling, have to be awaited for definite conclusions. Additional biomarker studies, as well as treatment combinations with, for example, ICI, could further improve the outcomes of this treatment strategy. 

## Figures and Tables

**Figure 1 vaccines-09-00525-f001:**
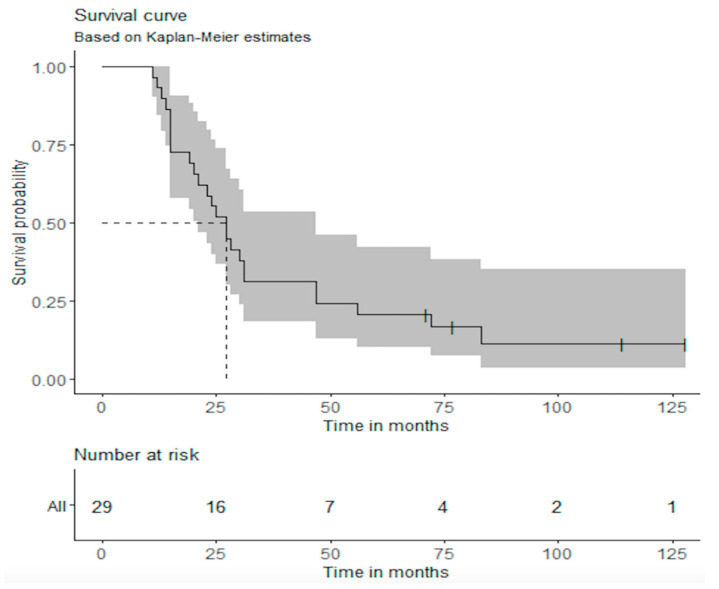
Overall survival of the combined phase I/II trials.

**Table 1 vaccines-09-00525-t001:** Patient characteristics.

Study	Gender	Age	Histology	Extended P/D	Chemotherapy	Response on DC	Survival (Months)	Alive
MM01 (*N*= 10)								
	male	68	epithelial	no	yes	PR	23	no
	male	63	epithelial	no	yes	PD	72	no
	male	55	epithelial	no	yes	PR	19	no
	male	66	epithelial	no	yes	PR	30	no
	male	71	epithelial	no	yes	SD	15	no
	male	64	epithelial	no	yes	PD	13	no
	male	75	epithelial	no	yes	PD	11	no
	male	77	epithelial	no	yes	PD	15	no
	male	70	epithelial	no	yes	PD	15	no
	male	58	epithelial	no	yes	PD	15	no
MM02 (*N* = 10)								
	male	62	epithelial	no	yes	SD	24	no
	male	71	epithelial	no	yes	SD	25	no
	male	78	epithelial	no	yes	SD	14	no
	male	55	epithelial	no	yes	CR	114	yes
	male	75	epithelial	no	yes	SD	27	no
	male	63	epithelial	yes	yes	SD	20	no
	male	58	biphasic	yes	yes	PD	12	no
	female	35	epithelial	yes	yes	PR	128	yes
	female	55	biphasic	yes	yes	SD	56	no
	male	48	epithelial	yes	yes	SD	83	no
MM03 (*N* = 9)								
	male	79	epithelial	no	no	SD	47	no
	male	69	epithelial	no	no	SD	31	no
	male	44	epithelial	no	yes	SD	77	yes
	female	59	epithelial	no	yes	PR	47	no
	male	73	epithelial	no	no	PR	71	yes
	male	67	epithelial	no	yes	SD	28	no
	male	68	epithelial	no	no	SD	21	no
	male	71	epithelial	no	yes	SD	31	no
	male	60	epithelial	no	yes	SD	27	no

The survival of all 29 patients is shown in Figure 1.

**Table 2 vaccines-09-00525-t002:** Overall survival analysis based on the Kaplan–Meier curve.

Study	Median OS (95% CI)	OS—2 Years (95% CI)	OS—5 Years (95% CI)
Overall	27 months (21–47)	55.2% (39.7%–76.6%)	20.7% (10.1%–42.2%)
MM01	15 months (15–Inf)	20.0% (5.8%–69.1%)	10.0% (1.6%–64.2%)
MM02	26 months (20–Inf)	60.0% (36.2–99.5%)	30.0% (11.6%–77.3%)
MM03	31 months (28–Inf)	88.9% (70.6%–100%)	22.2% (6.6%–75.4%)

**Table 3 vaccines-09-00525-t003:** Compiled toxicities of MM01–MM02–MM03.

Toxicity (*N* = 29)	Any Grade, *n* (%)	Grade 3–4, *n* (%)
Any AE	29 (100)	0
Injection site reaction	29 (100)	0
Fever	21 (72)	0
Dyspnea	8 (28)	0
Lab abnormalities	8 (28)	0
Gastrointestinal	8 (28)	0
Rash	3 (10)	0
Lethargia	3 (10)	0
Depression	1 (3)	0
Cardiomyopathy	1 (3)	1

## Data Availability

The data are not publicly available.

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
