# Peer review of "Long-Term Follow-Up of Mesothelioma Patients Treated with Dendritic Cell Therapy in Three Phase I/II Trials"

_vaccines, 2021, doi:10.3390/vaccines9050525_

Round 1

Reviewer 1 Report

The article presents a compilation analysis of three independent clinical studies conducted and previously published by the authors (manuscript’s references 13-15). The study emphasizes the potential of DC vaccination as a new therapeutic avenue to extend the survival of malignant pleural mesothelioma (MPM), currently considered an incurable disease. It is elegantly presented, well written and fits well within the journal’s scope. However, the novelty and importance of this work is questionable and some major points could be addressed to increase both the originality and impact of the study:

First, the authors hypothesize that ‘treatment with activated dendritic cells could preclude this T-cell exhaustion and therefore increase long term survival’. One of the main questions remains unanswered yet. This work clearly shows that DC treatment extended the OS of MPM patients, but there is no compiled data from the three studies on whether DC treatment precludes T-cell exhaustion or increases T-cell infiltration. The novelty and impact of the article would benefit from, for example, a comparison of T cell infiltrate in biopsies pre- vs post-DC treatment from patients. Does DC vaccination turn an immune-desert tumor into a hot tumor? Does it increase T-cell or reduce immunosuppressive cells infiltration?

Second, as discussed by the authors, the primary outcome of the 3 trials were safety and toxicity. Therefore, a compiled dataset of the 3 studies for toxicity adverse events encountered should be included (for example, in Table 1).

Third, in study 1, despite showing the benefit of DC treatment, the OS and clinical benefit rate is clearly lower than studies 2 and 3. Further insight in this point would greatly aid to the significance of the article.

Fourth, the authors nicely discuss potential impact of tumor load on DC therapy effectiveness. However, impact of other independent predictive markers of response to immunotherapy in solid tumors, such as tumor mutational burden (Marabelle et al., Lancet Oncol. 2020 Oct;21(10):1353-1365. doi: 10.1016/S1470-2045(20)30445-9), is not discussed and could also explain the differences between the studies 1 and 2. Although tumor mutational load is generally low in MPM, is the vaccination more effective if DCs are activated with lysates from tumors with high mutational load? Does high mutational load correlate with response or increased T cell tumor infiltration after DC treatment in studies 1 and 2 (activated with autologous tumor lysates)?

And fifth, the authors claim that some patients were subsequently treated with immune checkpoint inhibitors (ICI) in which DC vaccination could have improved ICI’s efficacy. They state that ‘in fact, eight of nine patients in the last trial were treated with immune checkpoint inhibition (ICI) therapy at some point in time’. The significance of the article would profit from further analysis of the OS of these “DC-vaccinated” patients in comparison to the OS of MPM patients treated with ICI (e.g. PROMISE-MESO trial, reference 3 of the manuscript). Does DC vaccination extend the response to ICI, e.g. higher OS, longer ICI-treatment response duration? Does prior treatment with DCs correlate with more severe AE derived from ICI?

Finally, some minor points:

Check spelling. For example, in the discussion, line 6, “T-cel” should be “T-cell”.

Adjust reference to journal’s format and language (some words are in Dutch)

Author Response

Long-term follow up of mesothelioma patients treated with dendritic cell therapy in three phase I trials

We thank the reviewers for the valuable comments that helped to improve the content of the manuscript. We have taken notice of the reviewers’ comments and have revised our manuscript in accordance with the given suggestions.

Reviewer 1:
The article presents a compilation analysis of three independent clinical studies conducted and previously published by the authors (manuscript’s references 13-15). The study emphasizes the potential of DC vaccination as a new therapeutic avenue to extend the survival of malignant pleural mesothelioma (MPM), currently considered an incurable disease. It is elegantly presented, well written and fits well within the journal’s scope. However, the novelty and importance of this work is questionable and some major points could be addressed to increase both the originality and impact of the study:

1) First, the authors hypothesize that ‘treatment with activated dendritic cells could preclude this T-cell exhaustion and therefore increase long term survival’. One of the main questions remains unanswered yet. This work clearly shows that DC treatment extended the OS of MPM patients, but there is no compiled data from the three studies on whether DC treatment precludes T-cell exhaustion or increases T-cell infiltration. The novelty and impact of the article would benefit from, for example, a comparison of T cell infiltrate in biopsies pre- vs post-DC treatment from patients. Does DC vaccination turn an immune-desert tumor into a hot tumor? Does it increase T-cell or reduce immunosuppressive cells infiltration?

Response:

We agree with the reviewer that this data would indeed increase the impact of our manuscript, however, sequential biopsies are not available for all but a few patients. Therefore, no tumor biopsy conclusions on T cell infiltrations can be drawn from our data. However, we determined T cell patterns in the peripheral blood before and after the DC-therapy and found that DC-therapy induces changes in T cell subsets. We added the reference of this analysis to the manuscript.

2) Second, as discussed by the authors, the primary outcome of the 3 trials were safety and toxicity. Therefore, a compiled dataset of the 3 studies for toxicity adverse events encountered should be included (for example, in Table 1).

Response:

Based on the reviewers comment, we added a paragraph regarding the toxicity to the manuscript and also added table 3 to show the compiled dataset on toxicity.

3) Third, in study 1, despite showing the benefit of DC treatment, the OS and clinical benefit rate is clearly lower than studies 2 and 3. Further insight in this point would greatly aid to the significance of the article.

Response:

We agree with the reviewer that numerically OS is lower in study one but again given the small dataset of each individual phase I/II study, we would not imply that study 2 and 3 have significantly better outcomes.

4) Fourth, the authors nicely discuss potential impact of tumor load on DC therapy effectiveness. However, impact of other independent predictive markers of response to immunotherapy in solid tumors, such as tumor mutational burden (Marabelle et al., Lancet Oncol. 2020 Oct;21(10):1353-1365. doi: 10.1016/S1470-2045(20)30445-9), is not discussed and could also explain the differences between the studies 1 and 2. Although tumor mutational load is generally low in MPM, is the vaccination more effective if DCs are activated with lysates from tumors with high mutational load? Does high mutational load correlate with response or increased T cell tumor infiltration after DC treatment in studies 1 and 2 (activated with autologous tumor lysates)?

Response:

We agree with the reviewer that this would be a very interesting subject for an additional study. However, TMB was not analyzed in our studies given the years that these studies were active. Nevertheless, we added the possible value of TMB as a biomarker to the discussion, and also added the reference.

5) And fifth, the authors claim that some patients were subsequently treated with immune checkpoint inhibitors (ICI) in which DC vaccination could have improved ICI’s efficacy. They state that ‘in fact, eight of nine patients in the last trial were treated with immune checkpoint inhibition (ICI) therapy at some point in time’. The significance of the article would profit from further analysis of the OS of these “DC-vaccinated” patients in comparison to the OS of MPM patients treated with ICI (e.g. PROMISE-MESO trial, reference 3 of the manuscript). Does DC vaccination extend the response to ICI, e.g. higher OS, longer ICI-treatment response duration? Does prior treatment with DCs correlate with more severe AE derived from ICI?

Response:

We agree with the reviewer that ICI treatment after DC vaccination is a very interesting subject for further research. Especially the added efficacy of ICI after activation of T-cells due to DC therapy could be of major clinical value. We are currently analyzing our data that we have on these patients and will publish this separately.

Reviewer 2 Report

I was happy to review the manuscript titled "Long-term follow up of mesothelioma patients treated with dendritic cell therapy in three phase I trials" by Prof Aerts' group, who have established themselves as experts in DC therapy in mesothelioma.

It is a concise and to-the-point manuscript describing summarised follow up data from three separate trials. The only comment I would make is that these follow up results look fairly promising and, as the authors comment, data from the phase II/III ongoing trial would be to conclude the value of DC vaccination therapy in this disease.

Author Response

Long-term follow up of mesothelioma patients treated with dendritic cell therapy in three phase I trials

We thank the reviewers for the valuable comments that helped to improve the content of the manuscript. We have taken notice of the reviewers’ comments and have revised our manuscript in accordance with the given suggestions.

I was happy to review the manuscript titled "Long-term follow up of mesothelioma patients treated with dendritic cell therapy in three phase I trials" by Prof Aerts' group, who have established themselves as experts in DC therapy in mesothelioma.

It is a concise and to-the-point manuscript describing summarised follow up data from three separate trials. The only comment I would make is that these follow up results look fairly promising and, as the authors comment, data from the phase II/III ongoing trial would be to conclude the value of DC vaccination therapy in this disease.

Response:

We thank the reviewer for the nice comment, and agree that the phase II/III study must give more insight.

Reviewer 3 Report

There are several limitations to this study considering the selected outcome, differences between trial's setup and no control group comparison.

The sample size is very low and differences in tretment increase the variability and the inconsistency of the results.

Statistical analyses is quite null and no boost in clinical reevance was performed.

Introduction explains a set of a-specific articles and not include potential input that may increase the effectiveness of results as Omics data (epigenetics, Gene expression, metabolomics, proteomics and other).

The outcome is very high level and general and clinical informations may be improved to better charachterize group history.

Results may be improved and discussion must be more specific to the outcomes under the study.

Abstract is also not related to the performed insight. Eacj consideration about the functiona aspects may be referenced.

In this stage, considering several limits, the work can not be considered for publication.

Editing are mandatory.

Author Response

Long-term follow up of mesothelioma patients treated with dendritic cell therapy in three phase I trials

We thank the reviewers for the valuable comments that helped to improve the content of the manuscript. We have taken notice of the reviewers’ comments and have revised our manuscript in accordance with the given suggestions.

There are several limitations to this study considering the selected outcome, differences between trial's setup and no control group comparison.

The sample size is very low and differences in tretment increase the variability and the inconsistency of the results.

Statistical analyses is quite null and no boost in clinical reevance was performed.

Introduction explains a set of a-specific articles and not include potential input that may increase the effectiveness of results as Omics data (epigenetics, Gene expression, metabolomics, proteomics and other).

The outcome is very high level and general and clinical informations may be improved to better charachterize group history.

Results may be improved and discussion must be more specific to the outcomes under the study.

Abstract is also not related to the performed insight. Eacj consideration about the functiona aspects may be referenced.

In this stage, considering several limits, the work can not be considered for publication.

Editing are mandatory.

Response:

We thank the reviewer for the comments. We agree that sample size is limited and that this has influence on the impact. We are currently performing a phase III study which will give more insight, however we do still think that the outcome of the patients here is still promising.  We do not think that there are inconsistencies in outcome in the 3 studies. Inclusion of different omics is of interest but beyond the scope of this paper. Statistics are simple but this is also warranted given the small sample size.

We have again looked at the results and the discussion and have adapted some points.

Round 2

Reviewer 1 Report

The authors have answered all my comments from the previous review. Although some of my questions remain unsolved, due to lack of tumor material to analyze or because it is being included in a separate publication, the authors answered all of them. The article fits within the scope of the journal and contributes to increasing the relevance of DC vaccination for cancer treatment.

There are some minor spell checks (missing spaces, commas…) to be corrected yet.

Author Response

We thank the reviewer for his or her view that the comments are addressed

Reviewer 3 Report

Thank you for the revision. Now, quality is improved and the manuscript can be considered for publication.

Lastly, I suggest to add a peper that explains the relationship between survival and DNAm in MPM cases. "Cugliari, G.; Catalano, C.; Guarrera, S.; Allione, A.; Casalone, E.; Russo, A.; Grosso, F.; Ferrante, D.; Viberti, C.; Aspesi, A.; et al. DNA Methylation of FKBP5 as Predictor of Overall Survival in Malignant Pleural Mesothelioma. 2020, 21, 12(11), 3470."

Author Response

We thank the reviewer for his or her comment that the manuscript can be considered for publication.

DNA methylation is certainly an interesting field of research and we are monitoring the progress of several research groups worldwide. However, the relationship between DNA methylation and long-term outcome of DC therapy is subject for future research and is not something that we would like to elaborate on in the current manuscript.